# In Vivo Evaluation of DMSA-Coated Magnetic Nanoparticle Toxicity and Biodistribution in Rats: A Long-Term Follow-Up

**DOI:** 10.3390/nano12193513

**Published:** 2022-10-08

**Authors:** Fernanda Paulini, Aline R. M. Marangon, Carolina L. Azevedo, Juliana L. M. Brito, Marcelle S. Lemos, Marcelo H. Sousa, Fabiane H. Veiga-Souza, Paulo E. N. Souza, Carolina M. Lucci, Ricardo B. Azevedo

**Affiliations:** 1Department of Physiological Sciences, Institute of Biological Sciences, University of Brasilia, Brasilia 70910-900, Brazil; 2Autonomous Veterinarian, Brasilia 70000-000, Brazil; 3Faculty of Ceilandia, University of Brasilia, Brasilia 70910-900, Brazil; 4Department of Cell Biology, Laboratory of Protein Chemistry and Biochemistry, Institute of Biological Sciences, University of Brasilia, Brasilia 70910-900, Brazil; 5Laboratory of Electron Paramagnetic Resonance, Institute of Physics, University of Brasilia, Brasilia 70910-900, Brazil; 6Department of Genetics and Morphology, Institute of Biological Sciences, University of Brasilia, Brasilia 70910-900, Brazil

**Keywords:** nanotechnology, nanotoxicology, nanosafety, iron nanoparticles, IONPs, dimercaptosuccinic acid

## Abstract

This work presents a long-term follow-up (300 days) of rats after a single intravenous injection of DMSA-coated magnetite nanoparticles (DMSA-MNP). The animals were systematically evaluated by hematological, biochemical, and ultrasound examinations, monitoring the same animal over time. In addition, oxidative stress evaluation, DMSA-MNP biodistribution, computerized tomography for ex vivo organs, and histopathology analysis were performed at the end of the experiment period. Overall, DMSA-MNP administration did not cause serious damage to the rats’ health over the course of 300 days post-administration. All animals presented hematological parameters within the normal limits, and no alterations on serum creatinine, urea, ALT, and AST were related to DMSA-MNP administration. Liver and spleen showed no important alterations in any of the examinations. The kidneys of treated animals displayed intermittent pelvis dilation at ultrasound analysis, but without damage to the organ parenchyma after 300 days. The lungs of treated animals presented a light interalveolar septal thickening, but the animals did not present any clinical respiratory symptom. Nanoparticles were not detected in the vital organs of treated animals 300 days after administration. This work represents the first assessment of the long-term effects of DMSA-MNP and goes a step further on the safety of its use for biomedical applications.

## 1. Introduction

Iron oxide nanoparticles have been applied as a nanomaterial in biomedical applications, such as tumor treatment by magneto-hyperthermia [1], and as a contrast agent for magnetic resonance imaging [2]. In fact, the use of iron oxide nanoparticles for liver or spleen imaging in humans was approved by the US Food and Drug Administration (FDA) in 1996 [3], with this product now commercially available for biomedical applications. Studies demonstrated that iron oxide nanoparticles are a harmless material for intravenous injection when the dosage is controlled [4,5,6,7].

These magnetic nanoparticles (MNP) are typically functionalized by surface coating with organic molecules to become biocompatible, soluble, and stable in organic systems and lower their aggregation [8,9]. Surface-coating is an important element affecting MNP interaction with cells and in vivo distribution [10]. Dimercaptosuccinic acid (DMSA) is one of the most used molecules for MNP surface coating, resulting in stable ferrofluids in biological media [11,12,13]. DMSA alone, approved by the FDA in 1989, shows low toxicity in various biological systems [4,5,6,7,14,15] and constitutes a good chelating agent when used for the treatment of human heavy metal poisoning [16].

Therefore, DMSA-coated magnetic nanoparticles (DMSA-MNP) present promising biomedical applications and have been largely studied. In general, iron oxide nanoparticles coated with DMSA do not affect cell viability in vitro (for a review see [17]). However, one of the biggest concerns about the use of nanomaterials in biomedical procedures is the possible side effects they may have on living organisms. 

Previous studies showed that DMSA-MNP are preferentially driven towards the lungs when injected intravenously in mice. DMSA-MNP was found in high amounts in the lungs just 30 min after injection [4]. These DMSA-MNP aggregates first appear inside large blood vessels within the lungs (after 24 h), then in capillaries (after 48 h) and later into parenchyma cells (after 7 and 15 days), with some present in the cytoplasm of macrophages within the bronchiolar lumen [18]. The quantity of DMSA-MNP aggregates decreased 90 days after the injection [5]. The presence of DMSA-MNP in the lungs caused increased levels of interleukin-1 and interleukin-6 as early as 12 h after injection [4], triggering an inflammatory process. Mild lung parenchyma inflammation was observed up to 15 days after DMSA-MNP administration, with an apparent increased number of inflammatory cells spread all over the parenchyma (Day 30). However, the inflammatory process reduces as a function of time [5].

In pigs, DMSA-MNP was observed primarily in the liver, spleen, and lungs 5 h after I.V. injection [19]. In non-human primates, DMSA-MNP were preferentially distributed to the lungs, liver, and kidneys at 12 h, 30, and 90 days after a single intravenous injection (0.5 mg Fe/Kg of body weight), but in reducing quantities over time post-administration. However, no effects were observed regarding hematological values or kidney and liver function. Additionally, all organs maintained a histologically normal appearance, except for the liver, which showed a slight expansion of the space of Disse 90 days after the injection [6]. DMSA-MNP were detected inside hepatocytes and their mitochondria looked swollen 90 days after injection, but the authors considered that DMSA-MNP were well-tolerated by capuchin monkeys [20].

It has been proposed that the preferential accumulation of DMSA-MNP in the lungs affords a useful property that could be exploited for lung disease treatments, directing specific drugs to the lungs without promoting side effects in extrapulmonary organs [18]. Nevertheless, the biological effects of any new treatment approach must be extensively evaluated before medical and clinical applications can be put forward. Although several studies have already investigated the possible toxic effects of DMSA-MNP administration on living organisms, the majority were carried out over the short (hours to days) and medium term (up to 3 months), and long-term studies are scarce. In addition, in vivo experiments generally use different animals for the different times analyzed, with animals being euthanized at specific timepoints. Since individual responses cannot be discarded, it is important to evaluate the same animal over time.

This work presents a long-term follow-up (300 days) of rats that received a single intravenous injection of DMSA-MNP. The animals were systematically evaluated by hematological, biochemical, and ultrasound examinations, with non-invasive methods allowing the same animal to be monitored over time. In addition, reactive oxygen species (ROS) and nitric oxide production in the blood for oxidative stress evaluation, DMSA-MNP biodistribution using electronic magnetic resonance, and computerized tomography for ex vivo organ analyses were performed at the end of the experiment period. Thus, this work represents the first assessment of the long-term effects of DMSA-MNP and goes a step further by investigating the safety of its use for biomedical applications.

## 2. Materials and Methods

### 2.1. Sample Preparation and Characterization

To synthesize DMSA-coated magnetic nanoparticles (DMSA-MNP), magnetite nanoparticles (Fe_3_O_4_) were firstly synthesized by the aqueous coprecipitation of Fe^2+^ and Fe^3+^ with ammonia and thus oxidized to maghemite (g-Fe_2_O_3_) into acidic medium using O_2_, according to the procedure previously reported in the literature [21]. Surface functionalization of magnetic cores with DMSA was performed by adapting a previously described procedure [12]. Briefly, an aqueous DMSA solution (0.3 mol/L) was added to the g-Fe_2_O_3_ suspension at a DMSA/Fe molar ratio of ~10%, under agitation, at pH 3.5 for 12 h. The as-produced dispersion was subsequently dialyzed for 12 h against deionized water to eliminate the free DMSA from the bulk dispersion and the pH was adjusted to ~6.5. All procedures were performed under a sterile hood.

X-ray powder diffraction (XRD) data were collected using an X-ray Miniflex 600 diffractometer (Rigaku, Tokyo, Japan), with Cu-Kα radiation (λ = 1.541 Å), operating at 40 kV and 30 mA. The average diameter of the nanocrystalline domain was estimated using Scherrer’s equation [22]. A transmission electron microscope (Jeol 1011, Jeol, Tokyo, Japan) was utilized to assess average particle size and size dispersion. Atomic absorption spectrophotometry was conducted using a commercial PerkinElmer 5000 system (PerkinElmer, Shelton, CT, USA). The zeta potential and hydrodynamic diameter were determined using a ZetaSizer Nano ZS analyzer (Malvern Instruments, Malvern, UK).

### 2.2. Experimental Design

All animal procedures performed in this study were approved by the Animal Ethics Committee of the University of Brasilia (protocol nº 17607/2016). A total of 15 healthy female Wistar rats aged 8–12 weeks, weighing 156.1 ± 5.1 g, were involved in this study. The rats were housed in polypropylene cages with woodchips as bedding in a ventilated room (~24 °C) under a 12 h/12 h light/dark cycle, with ad libitum access to commercial food and tap water.

The animals were randomly divided into three groups: T5 Group (*n* = 5), each animal received DMSA-MNP solution at 5 mg Fe/kg of body weight; T0.5 Group (*n* = 5), each animal received DMSA-MNP solution at 0.5 mg Fe/kg of body weight, and the control group (*n* = 5), each animal received saline solution. For DMSA-MNP solution/saline administration, the animals were first anesthetized with ketamine (5 mg/kg, I.M.—Rhobifarma-Hortolândia, SP, Brazil) and xylazine (0.5 mg/kg, I.M.—União Química-Embu-Guaçu, SP, Brazil), administering a single intravenous bolus injection (100 µL) into the tail vein.

The animals were weighed weekly, and their behavior was observed daily for signs of health deterioration. Blood samples were collected on D0 (before injection), D15, D30, and every 30 days until the 10th month after treatment. Abdominal ultrasound analyses were performed in all animals at the end of the 4th, 6th, 7th, 8th, 9th, and 10th months after treatment.

All animals were euthanized ten months (300 days) after the injection by anesthetic overdose (ketamine and xylazine) and cardiac puncture. Blood was slowly drawn from the right ventricle using a 3-mL heparinized syringe and a 24 G needle for HbNO and ROS evaluation. The liver, kidneys, spleen, lungs, heart, and aorta were harvested for further evaluation.

### 2.3. Blood Sampling and Analysis

For blood collection, animals were anesthetized by ventilation with isoflurane (BioChimico^®^, Rio de Janeiro, RJ, Brazil) in pure oxygen, with blood drawn from the jugular vein. The following hematological parameters were evaluated using an automated hematology analyzer (Sysmex pocH—110 iV Diff TM, Kobe, Hyogo, Japan): red blood cell count (RBC), hemoglobin concentration (HB), hematocrit (HTC), mean corpuscular volume (MCV), mean corpuscular hemoglobin (MCH), mean cell hemoglobin concentration (MCHC), white blood cell count (WBC), lymphocyte number (LN) and percentage (LP), monocyte number (MN) and percentage (MP), and platelets (PLT). The biochemical parameters evaluated were: alanine aminotransferase (ALT) and aspartate aminotransferase (AST) for liver function, and urea and creatinine for kidney function, using specific assay kits (Labmax 100—Labtest^®^, Lagoa Santa, MG, Brazil) in a ChemWell-T biochemical autoanalyzer (LabTest, Lagoa Santa, MG, Brazil). Serum iron levels (μg/dL) were also measured.

### 2.4. Ultrasound Analysis

Ultrasound analysis was performed using a veterinary color Doppler ultrasound (Z5 vet, Mindray, Guangdong, China) with multi-frequency micro-convex (5 to 8 MHz) and linear (7.5 to 10 MHz) probes. The kidneys, spleen, stomach, liver, and urinary bladder were evaluated. Renal volume and spleen length/width were also measured. The resistivity index (RI) of kidney interlobar arteries and the splenic artery were measured using the color Doppler tool. Doppler images were performed using at least three consecutive waves of the arterial spectral trace. The RI was calculated automatically by the equipment after manual delimitation of the peak systolic velocity and final diastolic velocity. This procedure was repeated two more times and the average RI of each artery was obtained for each animal.

### 2.5. Detection of Reactive Oxygen Species (ROS) and Nitrosyl Hemoglobin (HbNO) in Blood and ROS in Tissue Samples by Electron Paramagnetic Resonance (EPR)

Blood was collected in heparinized tubes and divided into two aliquots—one for HbNO detection relating to the amount of bioavailable nitric oxide (NO) in the blood and the other for ROS detection.

HbNO was measured as previously described [23], with modifications. One milliliter of blood was transferred to 1 mL syringes, centrifuged (2500 rpm, 5 min), frozen in liquid nitrogen, and maintained at –80 °C until analyzed. The frozen sample was directly transferred to a liquid nitrogen-filled quartz finger Dewar, which was placed into the EPR resonator. The HbNO concentration was recorded using a Bruker X-band spectrometer (Bruker EMX plus, Bremen, Germany) with a high sensitivity cavity (Bruker ER 4119HS, Bremen, Germany). The EPR spectrometer settings were: 10 mW microwave power, 5 G amplitude modulation, 240 G sweep width, number of scans = 4, and 12 s sweep time. The amount of detected NO• was determined using the calibration curve for intensity of the EPR signal of erythrocytes treated with known sodium nitrite concentrations (0.1–10 mM) and Na_2_S_2_O_4_ (20 mg).

The ROS detection method was based on monitoring the formation of EPR-detectable nitroxide (CM•), from the reaction of ROS with CMH (1-hydroxy-3-methoxycarbonyl-2,2,5,5-tetramethylpyrrolidine). A CMH solution (400 µM) was prepared in Krebs Hepes buffer (KHB) in the presence of 25 μM deferoxamine (DF), 5 μM diethyldithiocarbamate (DETC), and heparin sodium, (100 IU/mL), at pH 7.4. Blood was immediately treated with CMH in a 1:1 ratio. The tubes were incubated under gentle shaking at 37 °C for 30 min. Subsequently, 50 µL of the solution was placed between two ice blocks (200 µL each) in a 1 mL de-capped syringe and snap-frozen in liquid nitrogen [24].

For the analysis of ROS in tissue sections, small pieces (approximately 2 × 2 × 2 mm^3^) of liver, kidney, spleen, and lung were placed in separate tubes and washed three times each with KHB. A 700 µL aliquot of working solution containing 200 µM CMH, 25 µM DF, 5 µM DETC, and heparin sodium (50 IU/mL) was added. The tubes were incubated under gentle shaking at 37 °C. After 60 min, 450 µL of the sample was transferred to a 1 mL de-capped syringe and snap-frozen in liquid nitrogen.

All samples were stored at −80 °C until EPR measurements were performed. The instrumental settings were: 2 mW microwave power, 5 G amplitude modulation, 100 kHz modulation frequency, and 200 G sweep width. The peak height, corresponding to the distance between the lowest and highest points in the first derivative spectrum, was used for signal detection. The EPR signal was calibrated using a standard calibration curve with a range of nitroxide radical standard concentrations (CP•, 0, 5, 10, 50, and 100 µM).

### 2.6. DMSA-MNP Detection and Quantification in Organs by Ferromagnetic Resonance (FMR)

To determine the biodistribution of the DMSA-MNP applied in the treatment, we performed ferromagnetic resonance (FMR) experiments using the same equipment used in the EPR experiments. For this purpose, liver, spleen, kidney, and lung samples were macerated and homogenized in distilled water using a T 10 basic ULTRA-TURRAX^®^ homogenizer (IKA^®^ Werke, Staufen, Germany) and lyophilized (L101, Liotop, São Carlos, SP, Brazil). Lyophilized samples were placed into hematocrit glass capillaries, ensuring the same filling factor for all samples. The spectra were collected at room temperature, 10 G modulation, 2 mW microwave power, and one scan. A calibration curve was constructed using diluted samples with known DMSA-MNP concentrations (0.375, 0.75, 1.5, and 2.94 µg/mL) to determine the DMSA-MNP concentration in the different organs. Corresponding FMR spectra were collected, and the peak-to-peak amplitude of water-diluted DMSA-MNP samples versus concentration are shown in Figure 1.

### 2.7. Organ Micro-Computed Tomography (Micro-CT) and Histological Analysis

The organs harvested after euthanasia (the liver, kidneys, spleen, and lungs) were fixed in 10% formaldehyde-buffered solution for 48 h. The organs were immersed in a solution of 70% ethanol and 1% iodine tincture (10%) in water for 24 h under refrigeration (4 °C) to obtain contrast. Organs were scanned using a Skyscan 1076 MicroCT (Skyscan, Aartselaar, Belgium) at 100 kV, 100 mA, 9 μm pixel size, with an Al 1 mm filter, rotation step of 0.3, and 1 μSv/h for radiation. Bi-dimensional reconstruction of the images was performed using the NRecon software (V 1.6.9, version 64 bit with GPU acceleration) and three-dimensional reconstruction was performed using the CTvox software (V 2.7, version 64 bit) and DataViewer (1.5.0 version 32 bit). All software used were developed by Skyscan (Kontich, Belgium). A phantom object (a 50 mL polypropylene tube filled with deionized water) was used for the calculation of standard x-ray attenuation units (Hounsfield units, HU) using the CTvox software.

For histology, all organs previously microCT-scanned, plus the heart and aorta, were dehydrated in ethanol (70–100%), clarified in xylene, and embedded in Paraplast (Sigma Aldrich, St. Louis, MO, USA). Sections (5 μm thick) were cut using a Leica microtome (Leica RM2125 RTS Rotary Microtome, Leica Biosystems, Wetzlar, Germany). Three sample sections were randomly chosen from each block and placed in glass slides. Section were then deparaffinized, hydrated with decreasing concentrations of ethanol to distilled water, and stained with hematoxylin and eosin (H&E). Slides were then dehydrated in a graded series of ethanol, cleared in xylene, and mounted with a coverslip and Entellan (Sigma Aldrich, St. Louis, MO, USA). Sections were analyzed under a light microscope (Axiophot, Zeiss, Aalen, Germany). Images were captured by a Moticam 2300 camera (Meyer Instruments, Houston, TX, USA) and the Axio Vision program 40v 4.6.1.0.

### 2.8. Statistical Analyses

Results were analyzed using the GraphPad Prism 7 software (San Diego, CA, USA). Data were tested for normality by the Shapiro-Wilk test. The variables were compared among groups using analysis of variance (ANOVA) and Tukey’s test. Differences were considered significant when *p* < 0.05.

## 3. Results

### 3.1. DMSA-MNP Characterization

As shown in Figure 2A, the DMSA-MNP sample powder XRD pattern revealed a cubic spinel (Fd3m) structure, which matches the maghemite reference pattern according to the International Centre for Diffraction Data (ICDD; PDF No. 39–1346). Moreover, the average crystalline diameter of the magnetic core, calculated from the 311 XRD peak broadening using Scherrer’s formula, was 11.5 nm. TEM analysis (80 kV) showed that the DMSA-MNPs were approximately spherical in shape (Figure 2B). A mean (±SD) diameter of 17.2 ± 4.3 nm was estimated from the associated size histogram (Figure 2C) of DMSA-coated NPs counted in the TEM images.

The atomic absorption spectrophotometry determined the iron concentration in the DMSA-MNP suspension as 14.7 mg Fe/mL and the estimated number of particles: 16.5 × 10^16^ nanoparticles/mL. The zeta potential of the nanoparticles in the DMSA-MNP sample was −39 mV (at pH = 6.48) indicating successful functionalization of maghemite NPs with DMSA molecules [25]. This high zeta potential value also implies increased kinetic stability of the suspension—it is generally assumed that zeta potentials higher than |30 mV| are necessary to achieve this condition in charged colloids [26]. The average hydrodynamic diameter value (113.2 nm, PdI 0.256) determined from dynamic light scattering—with larger sizes than those determined from TEM and XRD—also indicates the presence of adsorbed molecules on the surface of maghemite NPs and/or the presence of small aggregates within the magnetic fluid suspension.

### 3.2. Clinical, Hematological, and Biochemical Analysis

In general, all animals exhibited normal weight gain (Figure 3) and a healthy state throughout the study.

The minimum and maximum values of the hematological and biochemical analysis of all animals before any treatment (D0—*n* = 15) are presented in the first rows of Table 1 and Table 2, respectively. These values were considered the limits of normality for this study, against which the values for the analyzed parameters from each animal at each time-point were individually compared.

Table 1 summarizes the hematological data of the control and treated animals. In general, all animals presented hematological parameters within the limits on D0. In a few cases, specific parameters were altered, but without correlation with DMSA-MNP administration either at 0.5 or 5 mg Fe/kg. Complete data are provided in Appendix A.

The biochemical analyses results of the control and treated animals are presented in Table 2. Animals from all groups presented creatinine values within the limits considered normal (D0), with a few exceptions. The mean urea values were slightly elevated between D15 and D90 compared to the D0 values, although most individual values were within the limits of normality. From D210 onwards, mean urea values and most individual values were lower than those observed on D0, irrespective of the treatment group. The mean ALT values were also elevated between D15 and D90 compared to D0, albeit within the limits of normality considered for this study. The mean AST values remained within the limits of normality, although some individual values were altered (both higher and lower than the limits), but with no correlation with DMSA-MNP administration. Serum iron values presented low variation in all three groups throughout the experiment.

### 3.3. Ultrasonography Analysis

Ultrasonography examinations revealed normal topography and echogenicity of the liver, spleen, stomach, and urinary bladder for all control and treated animals throughout the experiment. Renal volume ranged from 0.49 to 0.98 cm^3^, with the higher values in the later months. Spleen length and width varied from 0.95 to 1.97 cm and 0.45 to 0.57 cm, respectively. The mean RI ranged from 0.36 to 0.46 for the kidney interlobar artery, and from 0.27 to 0.38 for the spleen artery. In general, there was no variation in data over time or between groups for the aforementioned parameters.

Although the kidneys presented normal topography and normal parenchyma echogenicity, a number of DMSA-MNP-treated animals presented alterations compatible with hydronephrosis (Figure 4A), from both the T0.5 and T5 groups at different timepoints (Figure 4C). At the 4th month post-administration, one animal from the T0.5 group presented hydronephrosis, which was recurrent at the 6th, 7th, and 9th months. At the 6th month, another animal from the T0.5 group presented said alteration, which was observed again at the 9th month evaluation of the same animal. In the same group, two other animals presented hydronephrosis at the 7th, 8th, and 10th month timepoints. In the T5 group, all animals presented hydronephrosis at some stage of the experiment. The finding was more consistent at 10 months when four of the five animals had hydronephrosis. However, only one animal showed this alteration chronically: at the 8th, 9th, and 10th month evaluations. After euthanasia, histopathology only confirmed pelvis dilation in one animal (Figure 4B). In the control group, only one animal presented hydronephrosis at the 8th month, and the finding was not observed in the subsequent months. No correlation was observed between hydronephrosis occurrence and renal volume increase.

### 3.4. HbNO and ROS Detection in Blood and Different Tissues

No differences were found in HbNO or ROS (measured by CM• concentration) in the blood of treated (T5 or T0.5) and non-treated (Control) animals (Figure 5A,B) ten months after DMSA-MNP administration. Moreover, no significant differences were observed among the control and treated groups in the amount of ROS in the kidneys, liver, and lungs (Figure 5C,D,F). A significant increase (*p* < 0.05) in ROS production was only observed in the spleen in the T5 group compared to the control group (Figure 5E).

### 3.5. DMSA-MNP Detection in the Organs

The ferromagnetic resonance (FMR) spectra analysis of treated animal samples (T0.5 and T5 groups) showed entirely different profiles from those expected for DMSA-MNP, visualized in Figure 6A. Thus, we can rule out the presence of nanoparticles in the tissue samples tested. Another important result was obtained by comparing the peak-to-peak amplitudes of transitions in g ≈ 2.0 for the control and treated samples (T0.5 and T5 groups), presented in Figure 6B–E. Even with no presence of DMSA-MNP FMR signal (shown in Figure 6A), the samples presented a transition at around 3500 G (g ≈ 2.0) due to endogenous iron oxides. Comparisons between the peak-to-peak results of the control and treatment groups did not show any statistical difference (Figure 6B–E).

### 3.6. Micro-CT Analysis

Microtomography enabled precise visualization of the kidneys, liver, lungs, and spleen (Figure 7), and allowed a 3D reconstruction of the organs, permitting high-resolution visualization of specific virtual sections in any chosen axis. This tool even enabled the visualization of details inside the organs in this study. No alterations were observed in any of the organs analyzed. Notably, hydronephrosis was not observed in the kidneys.

### 3.7. Histopathology

Light microscopy analyses confirmed changes in renal pelvis dilation after euthanasia in one animal (Figure 4B), however, no renal parenchyma damage was observed in the animals (Figure 8). Moreover, all animals presented liver, spleen, cardiac tissue, and aortic artery with normal histology 10 months after DMSA-MNP injection (Figure 8). In the lungs, a light interalveolar septal thickening was observed in all DMSA-MNP treated animals, while control group animals had normal lung tissue morphology (Figure 9). Nevertheless, none of the animals presented any clinical respiratory symptoms.

## 4. Discussion

This study constitutes the first long-term follow-up (300 days) study of rats that received DMSA-MNP intravenously, monitoring the same animals over time. The overall safety assessment of DMSA-MNP in the rats was obtained by the combination of hematological and serum biochemistry profiles, ultrasound examinations, histopathological analysis, micro-computed tomography, ROS, NO levels in the blood, and biodistribution of SPION. It is noteworthy that the first exams were taken from the animals before the nanoparticle injection, and in this case, the animals themselves are the control.

Given the tendency of transition metals such as iron to amplify oxidant damage, one of the main concerns pertaining to iron-based MNP administration is overloading the body with iron, which can lead to oxidative stress in different organs, especially in those with high steady-state O_2_- and H_2_O_2_ production such as the liver [27,28], causing cirrhosis, cancer, and other diseases.

Using the FMR approach, we did not find DMSA-MNP 300 days post-administration in the liver, spleen, kidneys, or lungs, even in the T5 group, which received the highest dose. Although our results cannot discard the presence of DMSA-MNP in other (non-analyzed) organs, it is plausible to consider that they were mostly metabolized by the animal’s body. It is known that iron is basically stored in the hepatic parenchyma and the reticulum endothelial system (RES). The RES is generally a safe site for iron, keeping it sequestered, even after rather high doses [29]. In fact, iron nanoparticles are taken up by Kupffer cells and macrophages of the spleen and lungs where they are subsequently degraded and eliminated/re-stored in the body via normal iron metabolic pathways [4,5,7,30]. These include ferritin production, an iron-storage protein mainly found in the liver, but also in small amounts in most tissues which can limit the availability of redox-active iron to a certain extent and suppress iron-catalyzed oxidant damage to cells and tissues [27].

This is one example of several homeostatic mechanisms that can avoid damage resulting from iron accumulation. However, when those mechanisms fail or the amount of iron exceeds the capacity of the organ to eliminate it, it can lead to increased ROS production that can in turn induce cirrhosis and other diseases [27]. However, our 300-day post-injection results also demonstrated no raise in ROS level in the blood or the other organs analyzed (including the liver), except for the spleen (at the highest concentration dose), which exhibited a slight increase in ROS concentration. Although this ROS increase in the spleen is not clear, it may have no biological importance if it remains within a certain range.

As previously stated, iron overload can trigger cirrhosis and other chronic diseases, mainly in the liver. The doses used in the present study were 0.5 and 5 mg Fe/kg body weight, which is equivalent to the recommended clinical dose for some approved magnetic nanoparticles (~0.5 mg/kg body weight [7,19]) and 10 times the recommended dose. Considering that the 5 mg Fe/kg dose injected in rats is the equivalent to 0.8 mg Fe/kg in humans, based on body surface area [31], the impacts caused by DMSA-MNP could present a potential safety issue and a high-risk factor in cirrhosis. However, ultrasound examinations of the abdominal organs revealed no alterations in the liver, spleen, urinary bladder, or stomach. It is noteworthy that previous studies [4,5] reported that DMSA-MNP accumulation was higher in the liver after the seventh day, reaching a value six times higher than that observed in the lungs within 30 days of intravenous MNP administration. The same DMSA-MNP was observed inside hepatocytes, portal space connective tissue, the central lobular vein, and around the bile ducts, but without morphological changes in the liver [20]. As the ultrasound evaluations throughout the 300 days did not reveal any liver alterations and no damage was observed in post-euthanasia histopathological analysis, we can say that these MNPs are safe for the liver in the long-term at the doses used.

In the present study, the only ultrasound examination finding was renal pelvis dilation (hydronephrosis), which, although intermittent, appeared in almost all treated animals at both the 0.5 and 5 mg Fe/Kg doses at some point. In monkeys, a considerable amount of DMSA-MNP was observed in the renal tubules at 12 h and 30 days after intravenous injection, and it was still detected after 90 days, albeit in smaller amounts [6], with no morphological alterations observed in the kidneys. It was hypothesized [32] that nanoparticles (in general) could be the nucleation sites for kidney stones. Although kidney stones were not observed in any animals in our study by ultrasound, Micro-CT, or histopathology examinations, it is possible that small uroliths might have been produced and naturally expelled, resulting in the transient hydronephrosis observed. Nevertheless, it is important to highlight that kidney parenchyma was normal in the histopathological analysis 300 days post-DMSA-MNP administration.

None of the organs (liver, spleen, kidneys, and lungs) in the present study demonstrated alterations when evaluated by computerized microtomography after 300 days, indicating that the morphological changes observed by the previous works when injecting DMSA-MNP are transient, not inducing fibrosis or other diseases in the organs. In addition to the specific evaluations along the timeline of this study, we compared the blood analysis results, including hematological and biochemical data. It is important to note that blood cell count and other hematological parameters are evaluated clinically considering a normal range [33,34], and variations within the limits may not have a biological significance. On the other hand, values outside normal ranges are diagnostic for several disorders, but must always be considered together with other exams including clinical evaluation [35]. In the present study, the treated animals presented no clinical signs, together with normal body weight and food consumption throughout the experiment.

In general, animals presented no alterations in hematological or serum biochemical examinations relating to the treatment. Previously, iron oxide (magnetite) nanoparticles coated with DMSA were reported to have little effect on blood cell counts up to 30 days after administration of a single 2.5 mg/kg bw dose in Wistar rats [7]. Only a slight increase in leukocyte counts was observed 24 h after the injection when compared to the control [7]. On the contrary, a decrease in leukocyte numbers (compared to the control) was observed in a monkey from 15 to 90 days after DMSA-coated iron oxide (maghemite) nanoparticles administration, although the values remained within the normal limits for the species [6].

Changes in liver enzymes, primarily AST and ALT, could reflect hepatocyte membrane integrity, hepatocyte or biliary epithelial necrosis, cholestasis, or induction phenomenon [36]. Regarding our results, the biochemical parameters analyzed (ALT, AST, urea, and creatinine) were elevated in several animals, especially between 15 and 90 days after injection, regardless of the group. Since the control group also had the same changes, it is possible to correlate these alterations to the short period between the blood draws. Thus, from the time the alterations in the biochemical parameters were noted, a fluid replacement procedure was initiated, meaning each animal received 2 mL of subcutaneous saline solution after each blood collection. This procedure coincided with the return of parameters to normal limits from day 120. Other biochemical alterations observed during the experimental period were random and could not be related to the administrations.

The only finding that could give rise to some concern was the lung histology data, with lungs from animals treated with DMSA-MNP presenting discrete septal thickening. It is worth highlighting that none of the animals presented any respiratory symptoms throughout the experimental period and were already old at their time of death. Besides, only a small area of the lung presented this increase in interalveolar thickness. Previous studies reported that intravenously injected DMSA-coated nanoparticles are preferentially driven towards the lungs where inflammation is induced [4,5,6,20]. These same studies showed that the inflammatory process reduces as a function of time, with no additional pathologies, such as pulmonary fibrosis, observed. However, the aforementioned investigations did not exceed 90 days. In the present study, the lungs of the treated animals did not have inflammatory aggregates, only thickening of interalveolar septum, and MNPs were not detected in the lungs. Our findings, together with those of the other authors, suggest that DMSA-MNP were likely directed to the lungs soon after administration, where they may have triggered an inflammatory process (without symptoms) which resolved naturally, leaving a consequent interalveolar septal thickening that did not clinically compromise the animals’ breathing.

## 5. Conclusions

Overall, intravenous DMSA-MNP administration did not cause serious damage to the rats’ health over the course of 300 days after administration. Both the liver and spleen showed no important alterations in any of the examinations. The kidneys of treated animals, especially those which received 5 mg Fe/Kg, displayed intermittent pelvis dilation at ultrasound analysis, but without damage to the organ parenchyma after 300 days. Therefore, we can affirm that these MNPs are safe for these organs in the long term at the doses administered. However, the pulmonary effects must be further evaluated and any possible use of DMSA-MNP must consider this consequence.

## Figures and Tables

**Figure 1 nanomaterials-12-03513-f001:**
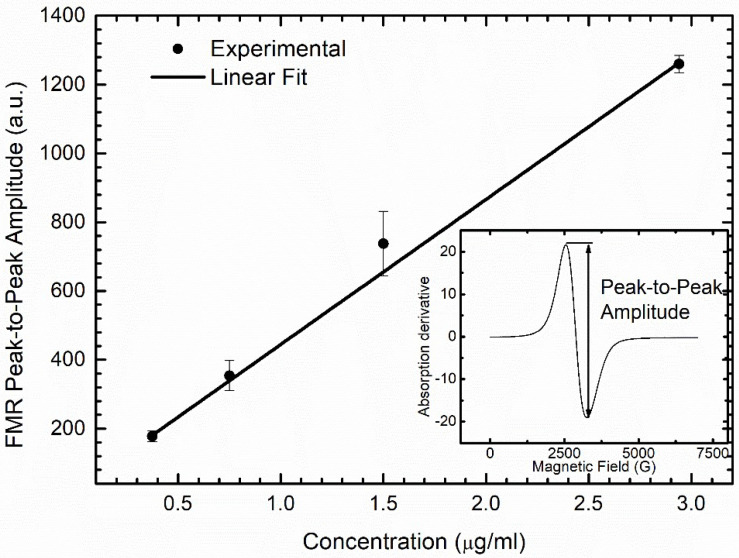
Results of peak-to-peak amplitude of FMR spectrum of water-diluted DMSA-MNP samples versus concentration. The linear equation obtained by fitting the peak-to-peak amplitude Ap-p, as a function of concentration C in (µg/mL) was Ap-p = 22.8 + 421.9C.

**Figure 2 nanomaterials-12-03513-f002:**
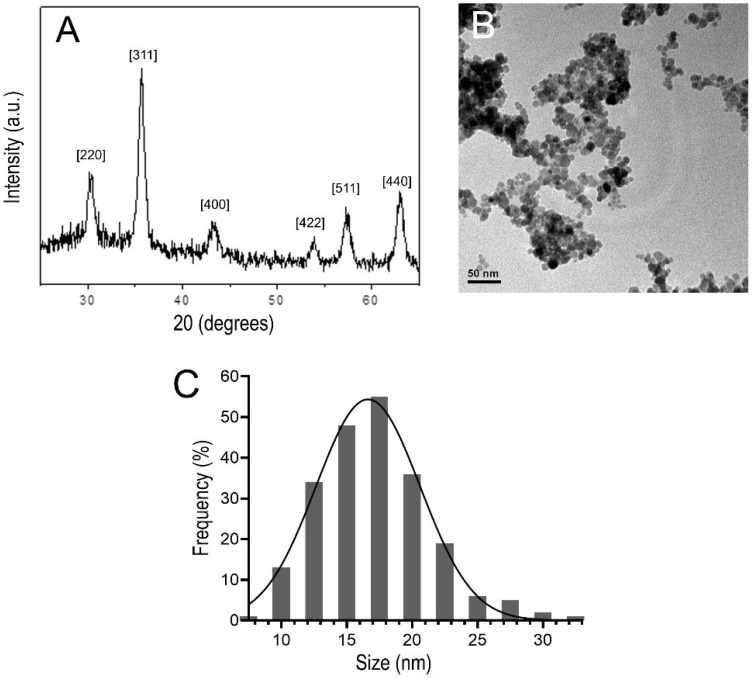
(**A**) X-ray powder diffraction pattern of DMSA-MNP, in the angular range from 20° to 70°. (**B**) Typical transmission electron micrograph of DMSA-MNP. (**C**) Nanoparticle size distribution histogram in DMSA-MNP.

**Figure 3 nanomaterials-12-03513-f003:**
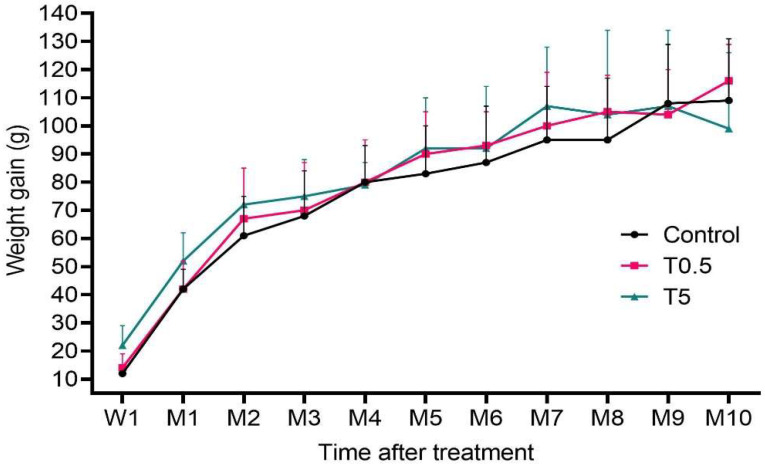
Mean (±SD) weight gain (g) per group of animals after the first week (W1) and over 10 months (M) of the experiment. There was no significant difference among groups (*p* > 0.05).

**Figure 4 nanomaterials-12-03513-f004:**
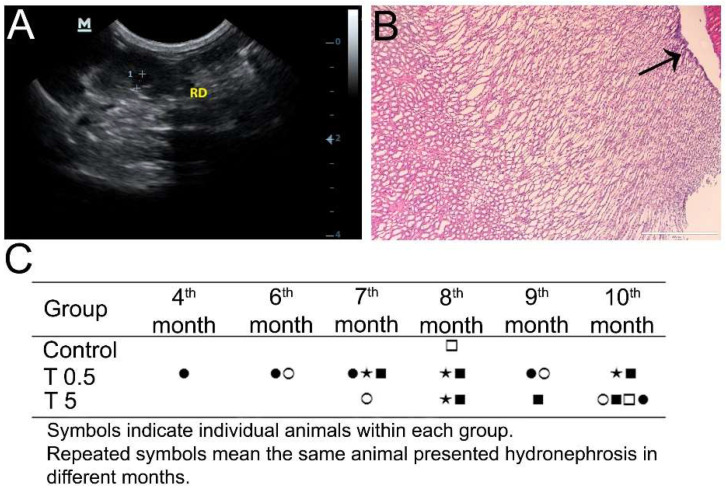
(**A**) Ultrasonographic image of a T0.5 Group animal kidney, with hydronephrosis (dilated pelvis shown between the two crosses). (**B**) Histopathological image of a T5 Group animal kidney showing a finding consistent with hydronephrosis (arrow shows the dilated pelvis), but without renal parenchyma damage (10× magnification, scale bar = 400 µm). (**C**) Hydronephrosis observed by ultrasound examination in experimental animals from both treated groups and the Control at different timepoints.

**Figure 5 nanomaterials-12-03513-f005:**
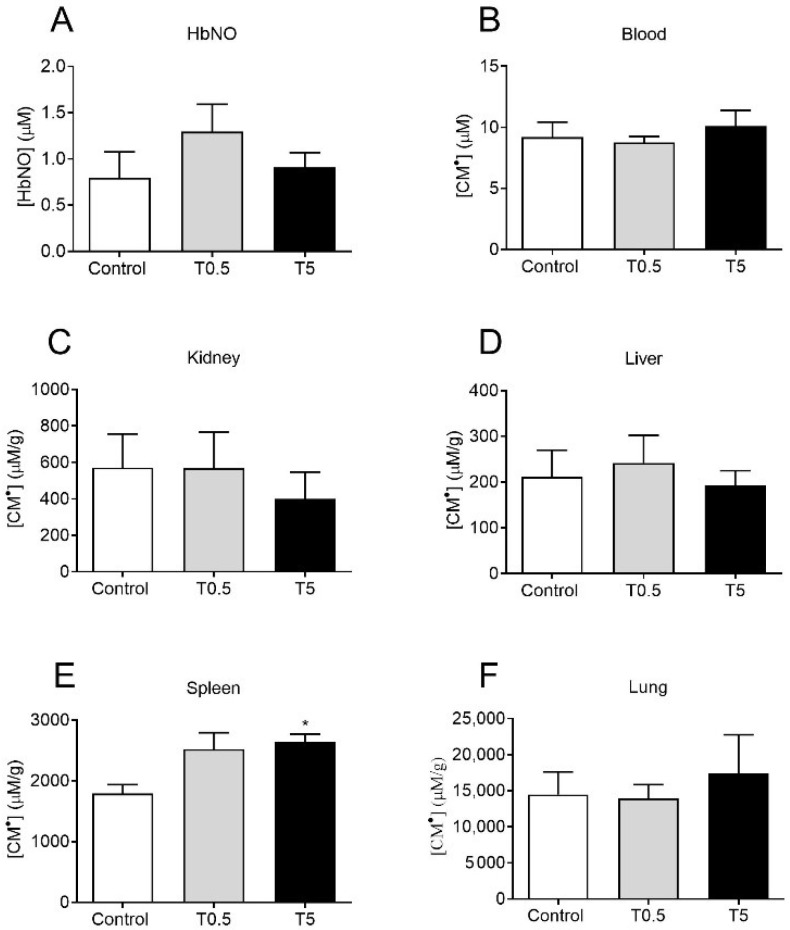
Quantification of (**A**) HbNO and (**B**–**F**) CM• in the blood, kidneys, liver, spleen, and lungs. The bars represent the concentration of HbNO and CM• in µM (blood) or µM/g for the different organs harvested 10 months after the administration of saline or DMSA-MNP (0.5 and 5 mg Fe/kg). The values represent the mean ± SEM. * *p* < 0.05, compared to the control group; one-way ANOVA, followed by Tukey’s multiple comparison test.

**Figure 6 nanomaterials-12-03513-f006:**
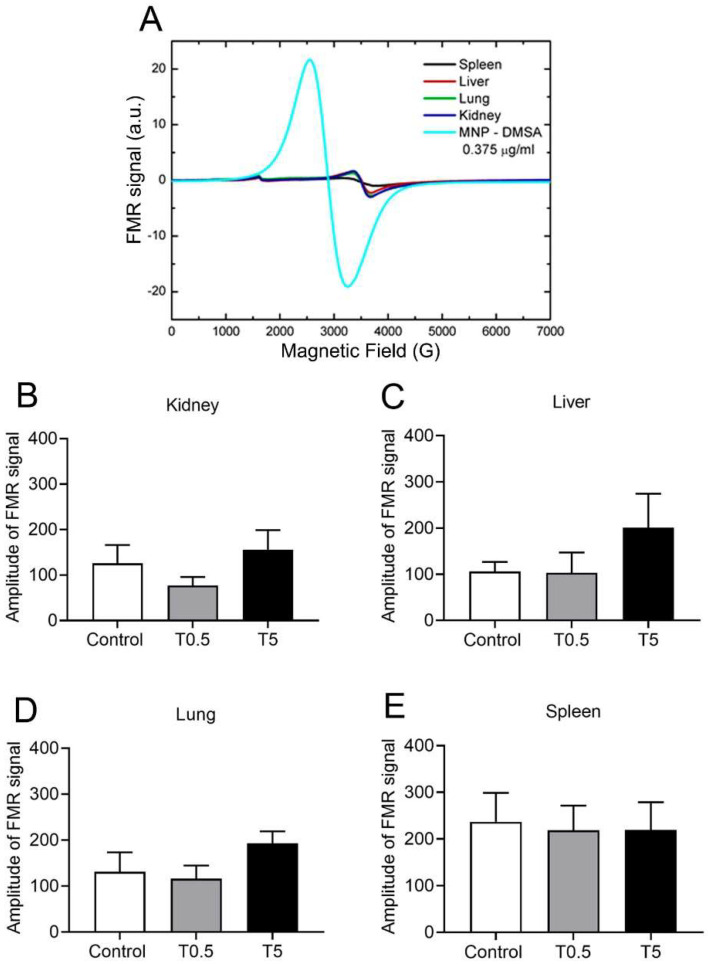
(**A**) Comparison of spectra of biological samples with pure DMSA-MNP spectrum. The tissue samples spectra presented the characteristic signal of Fe^3+^ high spin at 1500 G (g ≈ 4.3) was due to glass capillary contamination and another transition around 3500 G (g = 2.00) was due to endogenous iron oxide. The DMSA-MNP and tissue sample spectra are unmistakably different. (**B**–**E**) Peak-to-peak transition amplitude around 3500 G (g = 2.0) from kidney, liver, lung, and spleen samples in control and treatment conditions (groups T0.5 and T5). Data are expressed as mean ± SEM for 5 animals in each group.

**Figure 7 nanomaterials-12-03513-f007:**
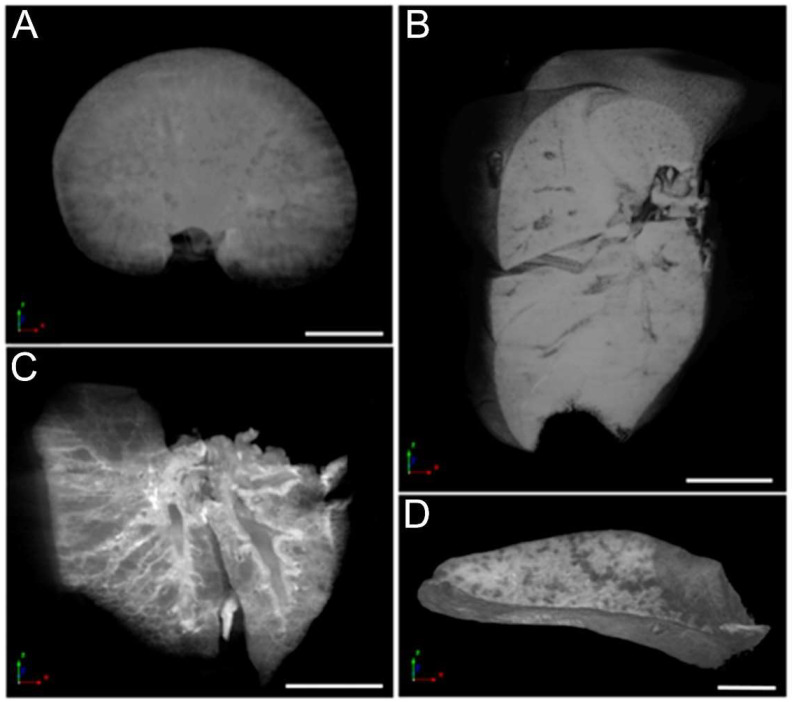
Micro-CT images of (**A**) kidneys, (**B**) liver, (**C**) lungs, and (**D**) spleen of a T5 Group animal. Scale bar = 2.5 mm.

**Figure 8 nanomaterials-12-03513-f008:**
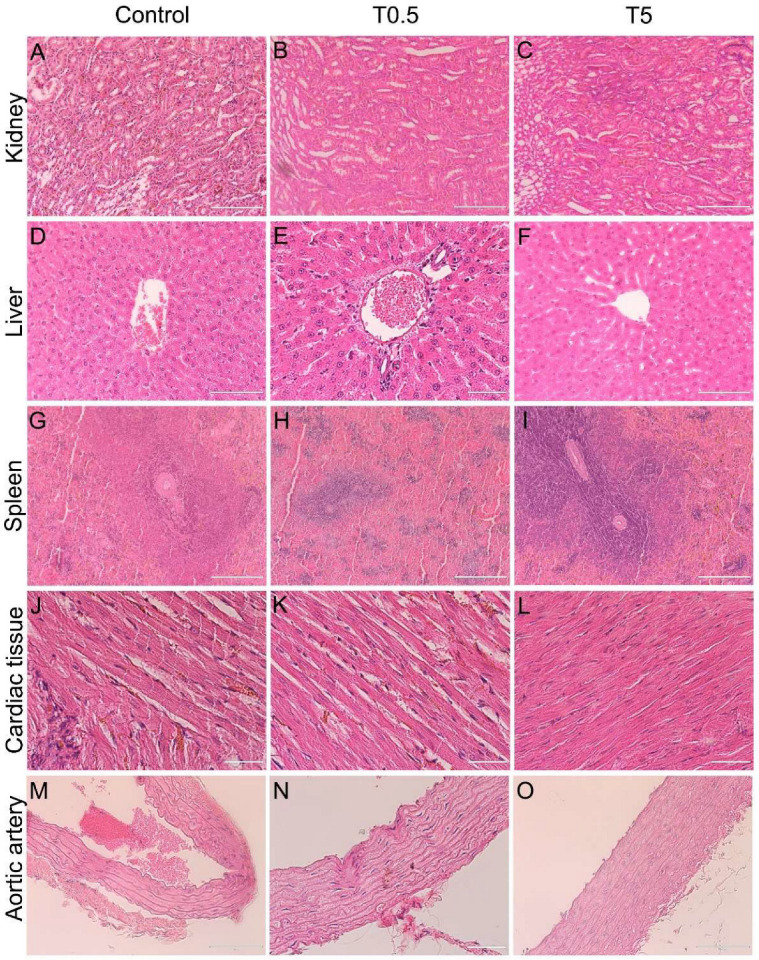
Representative micrographs of kidney, liver, spleen, cardiac tissue, and aortic artery in animals from the Control, T0.5, and T5 groups on Day 300 after the DMSA-MNP injection. All organs showed normal histological appearance. (**A**–**C**) Kidney (magnification 20×, scale bar = 200 µm), （**D**–**F**), liver (**D**,**F**)—40× magnification, scale bar = 100 µm; **E**—60× magnification, scale bar = 50 µm), (**G**–**I**) spleen (20× magnification; scale bar = 200 µm), (**J**–**L**) cardiac tissue (60× magnification; scale bar = 50 µm), and (**M**–**O**) aortic artery wall (**M**,**O**)—40× magnification, scale bar = 100 µm; (**N**)—60× magnification, scale bar = 50 µm).

**Figure 9 nanomaterials-12-03513-f009:**
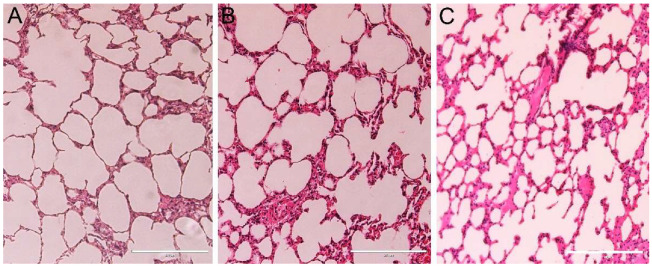
Representative micrographs of lungs from (**A**) the control group, presenting thin inter-alveolar septa, and the DMSA-MNP treated groups (**B**) T0.5 and (**C**) T5, showing discreet inter-alveolar septa thickening. 20× magnification. Scale bar = 200 µm.

**Table 1 nanomaterials-12-03513-t001:** Hematological data summary for control and treated animals. Complete data are provided in Appendix A.

		RBC (10^6^/µL)	HB (g/dL)	HTC (%)	WBC (10^3^/µL)	LP (%)	MP (%)
**D0** *		6.1–10.8	12.4–23.5	33.1–60.1	3.9–12.0	60.4–82.5	17.5–39.6
**Day after injection**	**Group**						
**D15**	Control	10.8 ± 1.5 (10.2–11.8)	22.2 ± 3.3 (20.8–24.7)	60.2 ± 8.8 (43.6–66.5)	8.9 ± 2.1 (6.2–12.2)	70.1 ± 3.9 (64.4–74.9)	29.9 ± 3.9 (25.1–33.1)
	T0.5	8.2 ± 1.8 (5.9–10.3)	17.5 ± 3.7 (12.4–22.3)	46.8 ± 9.7 (33.9–59.5)	8.9 ± 2.0 (6.1–11.7)	72.5 ± 6.1 (65.1–79.5)	27.5 ± 6.1 (23.8–34.9)
	T5	8.5 ± 1.2 (6.7–9.3)	18.3 ± 2.5 (14.2–20.4)	48.3 ± 6.2 (38.4–53.1)	7.9 ± 0.8 (6.8–8.7)	65.0 ± 8.2 (53.2–71.2)	34.9 ± 8.2 (28.5–46.8)
**D30**	Control	7.8 ± 0.3 (7.3–8.1)	15.7 ± 0.7 (14.6–16.7)	44.1 ± 1.7 (41.9–45.8)	10.7 ± 1.6 (8.2–12.7)	74.1 ± 3.8 (68.5–79.7)	25.9 ± 3.8 (20.3–27.8)
	T0.5	8.3 ± 0.4 (7.8–8.7)	17.2 ± 1.1 (16.9–18.1)	46.9 ± 2.4 (43.3–49.0)	10.3 ± 2.2 (8.1–13.5)	73.7 ± 3.7 (67.5–76.7)	26.3 ± 3.7 (23.3–32.5)
	T5	8.9 ± 1.2 (7.5–10.6)	18.5 ± 2.3 (15.9–21.5	50.6 ± 6.0 (44.4–58.8)	10.5 ±4.4 (5.2–15.5)	71.5 ± 11.8 (59.1–90.8)	28.5 ± 11.8 (9.2–40.9)
**D60**	Control	8.1 ± 0.2 (7.75–8.3)	16.0 ± 0.5 (15.5–16.6)	43.9 ± 0.8 (43.0–44.8)	8.9 ± 1.3 (7.7–10.7)	76.0 ± 2.1 (73.0–78.6)	24.0 ± 2.1 (21.4–27.0)
	T0.5	8.5 ± 0.7 (8.1–9.4)	17.4 ± 1.5 (16.0–18.4)	46.7 ± 3.7 (43.4–49.3)	9.3 ± 1.8 (6.8–11.1)	72.7 ± 2.4 (69.1–74.7)	27.3 ± 2.4 (24.9–28.2)
	T5	8.2 ± 0.3 (7.9–8.5)	16.6 ± 0.3 (16.2–16.9)	45.8 ± 1.0 (44.5–46.1)	6.3 ± 2.0 (4.7–9.0)	68.1 ± 6.0 (59.8–76.5)	31.9 ± 6.0 (23.5–40.2)
**D120**	Control	7.8 ± 0.2 (7.4–8.0)	15.4 ± 0.5 (14.7–16.0)	40.8 ± 1.8 (38.4–43.0)	5.7 ± 0.6 (4.8–6.6)	66.5 ± 7.4 (53.8–73.2)	33.5 ± 7.4 (26.8–46.2)
	T0.5	7.7 ± 0.4 (7.2–8.4)	15.7 ± 0.8 (14.4–16.7)	42.0 ± 1.9 (39.2–44.4)	5.2 ± 2.1 (3.4–8.3)	65.8 ± 4.5 (61.0–71.0)	34.2 ± 4.5 (29.0–37.7)
	T5	8.3 ± 0.7 (7.9–9.6)	16.6 ± 1.1 (15.9–18.5)	45.2 ± 3.3 (43.2–51.1)	5.3 ± 1.3 (4.5–7.5)	64.0 ± 6.8 (56.3–74.8)	36.0 ± 6.8 (25.2–43.7)
**D180**	Control	8.7 ± 1.2 (8.9–9.7)	17.0 ± 2.3 (13.0–18.6)	45.4 ± 6.1 (34.7–50.3)	5.1 ± 0.8 (3.6–5.5)	68.8 ± 1.9 (66.4–71.7)	31.2 ± 1.9 (28.3–33.6)
	T0.5	8.9 ± 1.3 (7.5–10.8)	17.7 ± 2.7 (15.2–21.5)	47.5 ± 6.8 (41.0–56.5)	4.9 ± 0.2 (4.0–7.0)	66.0 ± 4.6 (58.7–69.6)	34.0 ± 4.5 (30.4–41.3)
	T5	8.3 ± 0.4 (7.8–8.8)	16.6 ± 0.7 (16.2–17.3)	44.5 ± 2.0 (41.6–46.6)	3.7 ± 1.2 (2.9–5.8)	64.3 ± 5.2 (56.9–69.7)	35.7 ± 5.2 (30.3–43.1)
**D240**	Control	8.5 ± 0.5 (7.7–9.2)	17.0 ± 1.1 (15.6–18.4)	44.9 ± 2.2 (42.7–48.5)	5.5 ± 1.0 (4.4–6.9)	65.2 ± 2.9 (61.3–67.9)	34.8 ± 2.9 (32.1–38.7)
	T0.5	7.9 ± 0.6 (7.3–8.4)	16.1 ± 0.9 (15.0–16.9)	42.0 ± 3.2 (38.0–47.4)	6.1 ± 0.2 (5.9–6.3)	68.0 ± 8.5 (56.2–76.4)	32.1 ± 8.5 (23.6–43.8)
	T5	8.0 ± 0.6 (7.0–8.5)	16.3 ± 1.6 (13.8–17.9)	42.7 ± 4.0 (36.4–46.7)	4.5 ± 1.9 (2.9–7.6)	62.4 ± 4.9 (56.8–69.1)	37.6 ± 4.9 (30.9–43.2)
**D300**	Control	9.2 ± 1.3 (7.8–10.9)	18.0 ± 2.7 (15.3–21.8)	49.5 ± 7.6 (41.4–60.0)	3.2 ± 0.5 (2.4–8.6)	61.3 ± 5.5 (53.6–67.4)	38.7 ± 5.5 (32.6–46.4)
	T0.5	8.5 ± 0.7 (7.7–9.3)	17.1 ± 1.2 (16.0–18.6)	47.1 ± 3.9 (43.1–51.4)	2.5 ± 1.0 (1.8–3.9)	59.3 ± 6.3 (53.0–66.7)	40.7 ± 6.3 (33.3–47.0)
	T5	7.8 ± 0.3 (7.7–10.9)	15.6 ± 0.8 (15.3–21.8)	42.9 ± 2.0 (41.1–60.0)	2.5 ± 0.2 (1.8–4.1)	53.6 ± 0.6 (51.3–68.7)	36.6 ± 0.7 (31.1–48.7)

RBC: red blood cell count, HB: hemoglobin concentration, HTC: hematocrit, WBC: white blood cell count, LP: lymphocyte percentage, MP: monocyte percentage. * Minimum and maximum values for the hematological analysis of all animals (*n* = 15) prior to treatment. These values were considered the limits of normality for this study.

**Table 2 nanomaterials-12-03513-t002:** Results of serum biochemical analyses from control and treated animals. Complete data are shown in Appendix A.

		Creatinine (mg/dL)	Urea (mg/dL)	ALT (U/L)	AST (U/L)	Iron (μg/dL)
**D0** *		0.5–1.0	48–95	28–90	127–333	184–556
**Day after injection**	**Group**					
**D15**	Control	0.7 ± 0.1 (0.6–0.9)	80.8 ± 9.8 (69–94)	83.0 ± 24.0 (53–126)	218.5 ± 51.0 (170–288)	314.0 ± 26.9 (283–328)
	T0.5	0.7 ± 0.1 (0.6–0.7)	76.1 ± 7.2 (66–84)	79.6 ± 16.5 (65–112)	147.2 ± 16.8 (124–174)	349.3 ± 39.7 (294–396)
	T5	0.7 ± 0.1 (0.6–0.8)	76.4 ± 12.3 (63–92)	76.6 ± 17.9 (62–107)	142.8 ± 29.7 (109–178)	479.3 ± 97.6 (376–570)
**D30**	Control	0.9 ± 0.3 (0.8–1.4)	94.7 ± 9.1 (78–103)	90.8 ± 18.7 (68–117)	263.3 ± 84.9 (173–379)	394.6 ± 43.8 (317–420)
	T0.5	0.7 ± 0.1 (0.5–0.8)	66.0 ± 30.3 (15–87)	78.8 ± 35.1 (51–125)	238.0 ± 107.7 (136–336)	319.5 ± 268.0 (130–509)
	T5	0.8 ± 0.1 (0.7–0.9)	102.8 ± 7.9 (93–107)	69.8 ± 30.7 (26–91)	305.5 ± 48.8 (271–340)	403.0 ± 38.2 (376–430)
**D60**	Control	1.1 ± 0.6 (0.8–2.1)	80.8 ± 8.0 (73–89)	85.0 ± 50.6 (46–173)	233.4 ± 85.0 (153–364)	308.6 ± 73.7 (191–379)
	T0.5	0.8 ± 0.1 (0.8–0.9)	80.4 ± 4.3 (75–86)	74.0 ± 7.6 (66–84)	146.4 ± 71.3 (51–251)	347.0 ± 30.1 (317–396)
	T5	0.8 ± 0.1 (0.8–0.9)	79.8 ± 5.1 (74–85)	65.3 ± 14.5 (53–86)	206.8 ± 114.9 (137–378)	345.5 ± 42.2 (294–386)
**D120**	Control	0.5 ± 0.1 (0.4–0.7)	56.4 ± 10.9 (39–69)	46.4 ± 25.5 (19–80)	186.0 ± 53.3 (130–243)	263.0 ± 109.5 (140–434)
	T0.5	0.4 ± 0.2 (0.1–0.6)	52.2 ± 13.7 (36–70)	70.8 ± 36.6 (51–136)	201.2 ± 140.8 (106–445)	236.4 ± 40.9 (167–273)
	T5	0.5 ± 0.1 (0.5–0.6)	71.0 ± 4.8 (65–78)	67.2 ± 11.5 (56–85)	197.4 ± 64.4 (119–271)	254.6 ± 14.8 (239–270)
**D180**	Control	0.5 ± 0.1 (0.5–0.6)	51.6 ± 6.5 (47–63)	53.2 ± 23.9 (14–71)	158.2 ± 63.5 (62–230)	262.6 ± 44.3 (191–306)
	T0.5	0.7 ± 0.1 (0.6–0.8)	57.6 ± 4.3 (53–63)	58.6 ± 10.0 (44–69)	164.2 ± 39.0 (128–226)	280.4 ± 39.9 (223–333)
	T5	0.6 ± 0.1 (0.5–0.7)	53.8 ± 4.3 (49–60)	55.6 ± 75.0 (49–66)	176.0 ± 54.8 (116–266)	271.4 ± 38.6 (212–310)
**D240**	Control	0.7 ± 0.1 (0.6–0.8)	44.5 ± 3.1 (42–48)	51.0 ± 28.2 (18–89)	154.8 ± 34.3 (127–210)	286.8 ± 51.5 (250–375)
	T0.5	0.8 ± 0.2 (0.5–1.0)	52.8 ± 8.8 (39–61)	60.2 ± 11.6 (43–71)	213.4 ± 111.0 (122–374)	240.2 ± 49.9 (163–295)
	T5	0.8 ± 0.1 (0.7–0.9)	47.6 ± 6.4 (44–59)	60.4 ± 14.5 (53–80)	129.8 ± 40.7 (101–201)	348.6 ± 85.5 (243–465)
**D300**	Control	0.3 ± 0.3 (0.1–0.7)	39.8 ± 10.1 (23–50)	42.8 ± 22.8 (10–71)	192.2 ± 50.3 (118–232)	296,6 ± 131.5 (142–478)
	T0.5	0.5 ± 0.1 (0.4–0.6)	42.0 ± 12.0 (28–57)	52.5 ± 14.6 (35–65)	201.3 ± 52.2 (133–260)	343.5 ± 103.0 (235–474)
	T5	0.5 ± 0.4 (0.1–0.9)	49.2 ± 9.3 (43–64)	62.2 ± 37.8 (39–129)	185.4 ± 49.9 (126–226)	303.8 ± 139.0 (205–542)

ALT: alanine aminotransferase; AST: aspartate aminotransferase. * Minimum and maximum values for the biochemical analysis of all animals (*n* = 15) before any treatment. These values were considered the limits of normality for this study.

## Data Availability

All data in this study can be requested from the corresponding authors.

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
