# Peer review of "In Vivo Evaluation of DMSA-Coated Magnetic Nanoparticle Toxicity and Biodistribution in Rats: A Long-Term Follow-Up"

_nanomaterials, 2022, doi:10.3390/nano12193513_

Round 1

Reviewer 1 Report

The paper entitled In vivo evaluation of DMSA-coated magnetic nanoparticle 2 toxicity and biodistribution in rats: a long-term follow-up presents a long-term follow-up (300 days) of rats after a single intravenous injection of 17 DMSA-coated magnetite nanoparticles (DMSA-MNP). The paper presents the importance of the nanomaterials' effects on animal and human tests.

The paper is well organized and structured and contains hematological, biochemical, and ultrasound examinations, monitoring the same animal over time. In addition, oxidative stress evaluations, DMSA-MNP biodistribution, computerized tomography for ex vivo organs, and histopathology analysis were performed at the end of the experiment period. Overall, DMSA-MNP administration did not cause serious damage to the rats’ health over the course of 300 days post-administration.

The conclusions are concise and relevant for readers.

My observation is the authors use too many auto-citations (11). I suggest reducing them and replacing them with similar papers.

I suggest the publication of this paper in the present version, after reducing the auto-citations.

Author Response

We thank Reviewer 1 for the comments.

Point 1: My observation is the authors use too many auto-citations (11). I suggest reducing them and replacing them with similar papers.

I suggest the publication of this paper in the present version, after reducing the auto-citations.

Response 1:

We revised the text, which originally had 9 cited references that were from our research group, and decided to exclude 2 of them and include one from another research group (please, see lines 67-68).

Our research group has been working with DMSA-coated iron oxide nanoparticles since 1998 and is the group that has more publications on this subject. We only cited the papers we believe are important to contextualize and discuss the present results. The cited works followed a line of investigation, where the in vivo general effects of DMSA-MNP were evaluated in mice, in which we found the peculiar effects on the lungs and studied them in more detail (cited references 4, 5 and 18). We also evaluated the effects of the same MNP in non-human primates (cited references 6 and 20).

Along with these 5 references that are from our research group, one author of the present work co-authored 2 other papers in cooperation with another research group: a review article encompassing general aspects of development, characterization, and application in biomedicine of DMSA coated MNP, which is an important reference to base the present study (cited reference 17), and investigating the Hematotoxicity of MNP (cited reference 7), which is essential for the present work, since very few studies were performed evaluating effects on the blood counts.

Reviewer 2 Report

In this study, the authors prepared DMSA-MNP, administered it intravenously to rats, and then examined by hematology, biochemistry, and ultrasound for effects on vital organs to assess the safety of DMSA-MNP. The experiment results showed that DMSA-MNP did not cause severe health damage to rats after 300 days of administration. I think this is a well-done study with exciting findings. However, the paper needs minor revisions before it can be accepted for publication. My comments are as follows

1.       Further clarification is needed as to why the oxidative stress indicators were tested.

2.       I think the authors did not check the manuscript carefully, there are many punctuation errors, failure to properly superscript or subscript, and the manuscript is rough.

3.       The fifth keyword is not appropriate. Please revise it.

4.       Please standardize the font and color in the figure and label the magnification of the tissue specimen.

Author Response

We thank reviewer 2 for the comments. A point-by-point response to the reviewer’s comments is provided below.

Point 1: Further clarification is needed as to why the oxidative stress indicators were tested.

Response 1:

The excess of iron in the organism can trigger oxidative stress, which is a possible cause for organs’ damage. In previous work, signs of oxidative stress were observed after intravenous administration of oleic acid-pluronic-coated iron oxide MNPs in rats, more evidently in spleen and to a lesser degree in kidney and liver (Jain TK, et al. Biodistribution, clearance, and biocompatibility of iron oxide magnetic nanoparticles in rats. Mol Pharm. 2008;5(2):316–27).

Although our results did not show signs of organs damage, it was a possibility until the last approach (organs harvesting after animal’s death). If organs damage were to be found, one possibility was that it could be caused by oxidative stress, triggered by the excess of iron in the organism.

Point 2: I think the authors did not check the manuscript carefully, there are many punctuation errors, failure to properly superscript or subscript, and the manuscript is rough.

Response 2: Thank you for the observation. I suppose the errors occurred when copy-pasting the text to the template, which is not an excuse and we are truly sorry for that. The manuscript was thoroughly and carefully checked and the typing and spelling errors were corrected (these corrections are not highlighted on the text).

Point 3: The fifth keyword is not appropriate. Please revise it.

Response 3: Thank you for the observation. The fifth keyword was changed to IONPs. We also included Dimercaptosuccinic acid as a key-word (line 31).

Point 4: Please standardize the font and color in the figure and label the magnification of the tissue specimen.

Response 4: All Figures were replaced to standardize the font used in them. Magnification of tissue micrographs was added to the caption of Figures 4 (line 352), 8 (lines 402-408) and 9 (line 412).

Reviewer 3 Report

Comments on nanomaterials-1934196

In this manuscript, the authors studied the long-term effects of intravenously injected DMSA-coated magnetic nanoparticles in rats. This is an interesting topic to study. The authors investigated this topic with multiple imaging methods and biological assays. The manuscript was overall written well. However, the authors need to address several issues.

1.       How was the injection dose of DMSA-MNP determined? The authors need to provide a rationale.

2.       The experimental methods for tissue sectioning and staining should be added.

3.       Data were incompletely presented in some studies in this manuscript. For example, in Figure 4, the authors only studied the T0.5 group with ultrasonography. What about the control and T5 groups?

4.       The results of histological studies were not well-organized. The authors need to present all the H&E staining images for all the major organs collected from both the control and the experimental groups for comparisons.

5.       Standard deviations for data should be plotted in Figure 3.

Author Response

We thank reviewer 3 for the comments. A point-by-point response to the reviewer’s comments is provided below.

Point 1: How was the injection dose of DMSA-MNP determined? The authors need to provide a rationale.

Response 1: We determined the doses based on the recommended clinical dose of some approved magnetic nanoparticles, which is ~0.5 mg/kg body weight (according to Ruiz et al, 2015 and Edge et al., 2016). The dose of 0.5 mg/kg was also used in previous studies in non-human primates (Monge-Fuentes et al., 2011) and pigs (Edge et al., 2016). Since most substances become toxic at high doses, it is important to evaluate the nanotoxicity of high doses as well. So, we chose the clinical recommended dose (0.5 mg/kg body weight) and 10x it (5 mg/kg body weight) to evaluate the safety and possible side effects of the DMSA-MNP administration.

We included the following text on the discussion (lines 450-452): “The doses used in the present study were 0.5 and 5 mg Fe/kg body weight, which is equivalent to the recommended clinical dose for some approved magnetic nanoparticles (~0.5 mg/kg body weight [7, 19]) and 10 times the recommended dose.”

Point 2: The experimental methods for tissue sectioning and staining should be added.

Response 2: The methods for tissue sectioning and staining were better described in lines 232-238.

This part now reads: “Sections (5 μm thick) were cut using a Leica microtome (Leica RM2125 RTS Rotary Microtome, Leica Biosystems, Germany). Three sample sections were randomly chosen from each block and placed in glass slides. Section were then deparaffinized, hydrated with decreasing concentrations of ethanol to distilled water and stained with hematoxylin and eosin (H&E). Slides were then dehydrated in a graded series of ethanol, cleared in xylene and mounted with a coverslip and Entellan (Sigma Aldrich, St. Louis, USA).”

Point 3: Data were incompletely presented in some studies in this manuscript. For example, in Figure 4, the authors only studied the T0.5 group with ultrasonography. What about the control and T5 groups?

Response 3: We did study all animals from all groups by ultrasonography (please, see the description of the results in lines 325-334 and Figure 4C). We only selected a representative image to show in Figure 4A, which happened to be from a T0.5 group animal. We included that all animals were evaluated by ultrasonography in the material and methods to make it clearer. Lines 134-135 now read – “Abdominal ultrasound analyses were performed in all animals at the end of the 4th, 6th, 7th, 8th, 9th and 10th months after treatment.”

Point 4: The results of histological studies were not well-organized. The authors need to present all the H&E staining images for all the major organs collected from both the control and the experimental groups for comparisons.

Response 4: As requested, Figures 8 and 9 were replaced and now show all the major organs collected from both the control and the experimental groups.

Point 5: Standard deviations for data should be plotted in Figure 3.

Response 5: The Standard Deviation was included in Figure 3.

Round 2

Reviewer 3 Report

All my comments have been addressed. Therefore, I recommend acceptance of this manuscript.